# Extracellular Vesicle-Mediated Secretion of Protochlorophyllide in the Cyanobacterium *Leptolyngbya boryana*

**DOI:** 10.3390/plants11070910

**Published:** 2022-03-29

**Authors:** Kentaro Usui, Haruki Yamamoto, Takao Oi, Mitsutaka Taniguchi, Hitoshi Mori, Yuichi Fujita

**Affiliations:** Graduate School of Bioagricultural Sciences, Nagoya University, Nagoya 464-8601, Japan; usuikentarou@icloud.com (K.U.); haruki@agr.nagoya-u.ac.jp (H.Y.); oitaka@agr.nagoya-u.ac.jp (T.O.); taniguti@agr.nagoya-u.ac.jp (M.T.); morihito@agr.nagoya-u.ac.jp (H.M.)

**Keywords:** cyanobacteria, chlorophyll biosynthesis, protochlorophyllide, extracellular vesicles, porin, SLH/OprB-domain, outer membrane

## Abstract

Protochlorophyllide (Pchlide) reduction in the late stage of chlorophyll *a* (Chl) biosynthesis is catalyzed by two enzymes: light-dependent Pchlide oxidoreductase (LPOR) and dark-operative Pchlide oxidoreductase (DPOR). The differential operation of LPOR and DPOR enables a stable supply of Chl in response to changes in light conditions and environmental oxygen levels. When a DPOR-deficient mutant (YFC2) of the cyanobacterium *Leptolyngbya boryana* is grown heterotrophically in the dark, Pchlide accumulates in the cells and is secreted into the culture medium. In this study, we demonstrated the extracellular vesicle-mediated secretion of Pchlide. Pchlide fractions were isolated from the culture medium using sucrose density gradient centrifugation. Mass spectrometry analysis revealed that the Pchlide fractions contained porin isoforms, TolC, and FG-GAP repeat-containing protein, which are localized in the outer membrane. Transmission electron microscopy revealed extracellular vesicle-like structures in the vicinity of YFC2 cells and the Pchlide fractions. These findings suggested that the Pchlide secretion is mediated by extracellular vesicles in dark-grown YFC2 cells.

## 1. Introduction

Chlorophyll (Chl) is a tetrapyrrole pigment essential for photosynthesis. In cyanobacteria, Chl *a* is biosynthesized via 15 sequential enzymatic reactions that begin with glutamate [1,2], and the resulting Chl molecules eventually bind to Chl-binding proteins, such as PsaA/B proteins [3,4,5,6]. Protochlorophyllide (Pchlide) reduction, the penultimate reaction, in Chl *a* biosynthesis, is catalyzed by two enzymes with different evolutionary origins: light-dependent Pchlide reductase (LPOR) [7,8] and dark-operative Pchlide reductase (DPOR) [9,10] (Figure 1). Each of the two enzymes has distinct enzymatic properties, such as light requirement (LPOR) and oxygen sensitivity (DPOR), and they complement each other’s activity to ensure a stable supply of Chl in various environments, including light intensity [11] and oxygen levels [12]. Chl biosynthesis in cyanobacterial DPOR-deficient mutants stops at the Pchlide reduction in the dark, resulting in abnormal accumulation of Pchlide in the cells [13,14,15,16]. A significant portion of the accumulated Pchlide is secreted into the extracellular medium, turning the culture medium yellow. However, the mechanism by which Pchlide is secreted into the culture medium remains unknown.

Extracellular vesicles (EVs) are nanosized structures surrounded by a lipid bilayer derived mainly from the outer membranes, and all organisms can form EVs [17,18]. EVs are involved in various functions, such as cell-to-cell communication, biofilm formation, nutrient release/absorption, detoxification, horizontal gene transfer, metabolites secretion, and response to environmental stress. EV formation has been discovered in several cyanobacteria, including the model cyanobacterium *Synechocystis* sp. PCC 6803 [19,20]. The roles of cyanobacterial EVs have been suggested to include adaptive responses to various environmental stresses, defense against cyanophages, horizontal gene transfer, and biofilm formation [20]. However, the involvement of EVs in Chl biosynthesis is yet to be reported.

In this study, we used a *chlL*-disrupted mutant YFC2 of the filamentous cyanobacterium *Leptolyngbya boryana* strain dg5, which can grow heterotrophically in the dark [21]. Because the *chlL* gene encodes one of the DPOR subunits, Chl biosynthesis of YFC2 is stopped at the Pchlide reduction in the dark [13]. Pchlide accumulation and secretion of YFC2 were induced by heterotrophic culturing in the dark. The culture media containing Pchlide were analyzed using high-performance liquid chromatography (HPLC), sucrose density gradient centrifugation, and mass spectroscopy. The results suggested that Pchlide was secreted into the culture medium via EVs. Transmission electron microscopy (TEM) also revealed EV-like structures in the Pchlide fractions. These findings indicated EV-mediated Pchlide secretion, which accumulated in dark-grown YFC2 cells.

## 2. Results

### 2.1. Accumulation of Pchlide in Culture Media

YFC2 lacks *chlL*, which encodes a subunit of DPOR, and thus Chl biosynthesis in the dark is stopped at the Pchlide reduction stage, resulting in Pchlide accumulation in the cells [13,21]. In addition to Pchlide accumulation in the cells, a significant portion of Pchlide is secreted to the culture medium. First, we determined the Pchlide concentration in the cells and culture medium during dark heterotrophic growth (Figure 2). Pchlide accumulation became more pronounced as the cells proliferated and equal amounts were detected in the cell pellet and culture medium fractions up to 120 h (day 5), indicating that about half of the accumulated Pchlide was secreted into the culture medium. The Pchlide content of the culture medium peaked at about 35 nmol (in 10 mL) on day 5, then gradually decreased but remained above 20 nmol. Pchlide concentration in cell pellets continued to increase and reached about 130 nmol after 192 h (day 8). Chl content was nearly constant (20–40 nmol in 10-mL cultures), and no extracellular Chl was detected.

Pigments in the culture medium were analyzed using HPLC (Figure 3A, black trace). Monovinyl Pchlide was detected as the major pigment (peak a, Figure 3A,B), with a trace amount of protopheophorbide (Mg-depleted Pchlide; peak b, Figure 3A,C) also present. To confirm that this pigment is protopheophorbide, purified Pchlide was treated with hydrochloric acid to induce demetallation to form protopheophorbide. The absorption spectrum and elution time of HPLC confirmed that peak b was protopheophorbide (Appendix A).

The culture medium was subjected to ultracentrifugation to analyze the state of Pchlide within it. Pchlide was recovered almost entirely as a dark-green precipitate (Fraction 0). This precipitate was suspended in the buffer, and the pigments were analyzed using HPLC (Figure 3A, red trace). The elution profile was almost identical to that of the culture medium. The absorption spectra of the two peaks matched those from the culture medium (Figure 3B,C). This indicated that the pigments in the culture medium existed in a state where they could be precipitated by ultracentrifugation.

### 2.2. Fractionation of Pchlide in the Culture Medium

Fraction 0 was fractionated by sucrose density gradient ultracentrifugation (Figure 4A, inset) because EVs in diverse prokaryotes were fractionated by sucrose density gradient centrifugation [22]. Thus, the pigments were separated into a light-yellow band between 0.6 and 1.6 M sucrose (Fraction 1) and a dark-green band between 1.6 and 2.5 M sucrose (Fraction 2). Each fraction was collected and the absorption spectra of the fractions (Figure 4A) and their methanol extracts were recorded (Figure 4B). The main Soret peak at 418 nm, which is characteristic of protopheophorbide, was prominent and higher than the Soret peak of Pchlide at 434 nm in Fraction 1 (Figure 4B, trace 1). Fraction 2 had an absorption spectrum showing that monovinyl Pchlide was the main pigment (Figure 4B, trace 2), which was nearly identical to that of Fraction 0 (Figure 4B, trace 3). The pigment composition of the fractions was confirmed using HPLC analysis (Figure 4C).

Western blotting analysis using specific antisera for CP47 (PsbB) and CmpA as marker proteins in the thylakoid and plasma membranes, respectively, was used to evaluate the contamination of thylakoid and plasma membranes in the fractions (Figure 5). CP47 was not detected in any fractions, indicating very little thylakoid membrane contamination. This is consistent with the result that Chl was not detected in any fractions (Figure 4B,C). CmpA, the substrate-binding protein of bicarbonate transporter that localizes to the plasma membrane [23], was also not detected in any fractions. These results indicated the limited contamination of the plasma membranes.

SDS-PAGE was used to determine the protein composition of the fractions (Fractions 0, 1, and 2; Figure 6). In Fraction 0, six major bands a to f were clearly detected with apparent molecular masses of 78, 56, 53, 50, 32, and 29 kDa, respectively. These six proteins were further separated into Fractions 1 and 2 by sucrose density gradient centrifugation. Three proteins (bands a, e, and f with apparent molecular masses of 78, 32, and 29 kDa, respectively) were detected in Fraction 1 (lane 4), and three proteins (bands b, c, and d with apparent molecular masses of 56, 53, and 50 kDa, respectively) were detected in Fraction 2 (lane 5). These results indicate that the pigments (primarily monovinyl Pchlide) secreted into the culture medium are in two different states, which can be separated by the density difference, and that each pigment fraction contains specific sets of proteins.

### 2.3. Protein Composition of the Pchlide Fractions

To identify proteins of the fractions that interacted with Pchlide, each SDS-PAGE band was excised from the gel, trypsin-digested, and subjected to mass spectrometry. Table 1 lists the proteins that contained more than eight peptide fragments that matched with more than 95% accuracy based on the measured molecular mass values of the peptides and the genomic information of *L. boryana* (Appendix A).

The 78 kDa protein (band a) was identified as the FG-GAP repeat-containing protein encoded by LBDG_07530 and shared by Fractions 0 and 1. There appeared to be several overlapping proteins with almost the same mobility for the three proteins with molecular masses around 50 kDa in Fraction 0 (bands b, c, and d), and three different proteins were detected for each band. Porin (LBDG_40860), glutamine synthetase (LBDG_41490), and outer membrane efflux protein TolC (LBDG_23750) were detected in band b. Three porin isoforms (LBDG_25860, LBDG_40860, LBDG_41910) were detected in band c, while another set of three porin isoforms (LBDG_41910, LBDG_25860, LBDG_34630) were detected in band d. Two porin isoforms (LBDG_41910 and LBDG_25860) were commonly detected in bands c and d. These results indicate that four isoforms of porin are contained around 50 kDa in Fraction 0. The 31 and 29 kDa proteins (bands e and f) were not identified due to their low contents and lack of statistically significant peptide information.

Fraction 1 showed 78, 31 and 29 kDa proteins as the major proteins, with only 78 kDa identified as an FG-GAP repeat-containing protein, as in Fraction 0. Fraction 2, which is higher density and retains most Pchlide, has three bands around 50 kD as the major proteins, and three isoforms of porin were detected in common with Fraction 0.

FG-GAP repeat is a motif observed in extracellular proteins such as adhesins and integrins [24]. TolC, an efflux protein common to RND-type transporters and the type I secretion system (T1SS) [25,26], is localized to the outer membrane. Cyanobacterial porin, the SLH/OprB-domain-containing protein, is also localized to the outer membrane [20]. Thus, Fractions 1 and 2 contained mainly outer membrane-localized proteins, indicating that Pchlide is secreted via EVs probably derived from the outer membrane of YFC2.

### 2.4. Electron Microscopy of the Pchlide Fractions

We performed TEM to determine whether EVs are derived from outer membranes of YFC2 cells (Figure 7). dg5 (wild type) and YFC2 cells were cultured in the dark. YFC2 cells contained fewer and fragmented thylakoid membranes due to Chl deficiency caused by DPOR loss compared to wild type (Figure 7A,B), as observed in the previous report [27]. Interestingly, several EV-like structures were observed in the vicinity of YFC2 cells. We observed an interesting image in which the vesicles and the outer membrane of the cell appeared to be attached (red arrow in D). In this image, the vesicles membrane and the outer membrane appeared to form a continuous membrane structure. We observed 41 EV-like structures and measured their area. The diameter of the circle corresponding to their area was calculated, which ranged from a minimum of 25 nm to a maximum of 194 nm, with an average of 80 nm (Appendix A). The average size is close to that of EVs in a filamentous cyanobacterium *Cylindrospermopsis raciborskii* [28].

Fractions 0, 1, and 2 were also observed using TEM (Figure 8). A large number of single-membrane spherical vesicles that appeared to be EVs were observed in all fractions. Fraction 0 and 2 showed mainly single-layered spherical vesicles, with some multi-lamellar and pear-shaped vesicles. Such forms of EVs have also been observed in the EVs of other bacteria [29]. The number of EVs in Fraction 1 was less than that in Fractions 0 and 2 and were mostly single-membrane and spherical; however, multi-lamellar EVs were also observed at a lower frequency (Appendix A). The area of the EVs of Fractions 0, 1, and 2 was measured and the diameter of the circle corresponding to the area was calculated (Appendix A). The size distribution of Fraction 0 was most similar to that of EV observed in the vicinity of cells. EVs of Fractions 1 and 2 were slightly larger and slightly smaller than those of Fraction 0, respectively.

Based on the results of the analyses of the Pchlide-containing culture media, it is suggested that Pchlide accumulated in dark-grown YFC2 cells is secreted into the culture medium through EVs.

## 3. Discussion

In this study, we prepared Pchlide fractions from the culture media of dark-grown YFC2 cells and analyzed them using mass spectroscopy. Porin isoforms and TolC localized in the outer membrane were detected as the major proteins in Pchlide fractions. TEM detected EV-like structures in the vicinity of YFC2 cells and the Pchlide fractions. Considering that cyanobacterial EV is derived from the outer membrane, these results indicated that Pchlide accumulated in YFC2 cells is secreted into the culture medium via EVs.

When YFC2 was cultured in the dark, approximately half of the Pchlide concentration was secreted from the cells until day 5 (Figure 2). While the concentration of Pchlide in the culture medium did not increase after day 5, Pchlide concentration recovered in the cell pellet continued to increase. It was noted that a dark-green pellet, which differed from cells, was precipitated by low-speed centrifugation with cells after day 5 (Appendix A). The dark-green aggregates were difficult to separate from the cell pellet. A portion of the pellet was carefully collected using a spatula and analyzed. The pigment and protein compositions were very similar to those of Fraction 2 (Appendix A), suggesting that the dark-green pellet may be an EV aggregate. As the density of the EVs in the culture medium increases, EVs may aggregate with each other and precipitate. Therefore, it should be noted that after 5 days, the cell pellets comprised such EV aggregates containing Pchlide. The detailed analysis of the aggregates is a subject for future research.

The Pchlide fractions (Fractions 0, 1, and 2) from the culture supernatant were found to contain Pchlide and protopheophorbide as major pigments, as shown by the absorption spectra of the methanol extracts and HPLC analysis (Figure 4). The fractions contained very little Chl and CP47, indicating no substantial contamination of the thylakoid membrane (Figure 5). Interestingly, the comparison of the absorption spectrum of Fraction 2 with its methanol extract shows a significant shift in the Pchlide absorption peak toward longer wavelengths (Figure 4A,B). The Soret and Qy peaks are red-shifted by 40 nm and 47 nm, respectively. Such a red shift of Pchlide is observed in the photoactive LPOR-NADPH-Pchlide ternary complex (642 nm) [30]. However, the red shift observed in the present study is even larger than this. This suggests that Pchlide exists in an unusual aggregated state in EVs.

In the mass spectrometry analysis, glutamine synthetase (GlnA) was detected in Fraction 0 in addition to porin isoforms and TolC localized in the outer membrane. Proteomics of EVs in *E. coli* has identified over 100 proteins, in which GlnA was included together with porin and TolC [22]. The presence of GlnA in Fraction 0 may have some physiological significance rather than mere contamination of the cytoplasmic fraction.

Pchlide in the culture medium was recovered in Fraction 0 and separated into two fractions (Fractions 1 and 2) by sucrose density gradient ultracentrifugation. Most Pchlide was recovered in Fraction 2 with higher density, while protopheophorbide was mainly recovered in Fraction 1 with lower density. These fractions also differed in protein composition, with Fraction 2 having porin isoforms as major proteins, a feature similar to that of the outer membrane fraction of another cyanobacterium *Synechocystis* sp. PCC 6803 (two porin isoforms as major proteins) [20]. However, no genes encoding FG-GAP repeat-containing protein-homologs were discovered in the genome of *Synechocystis* sp. PCC 6803 or *E. coli*. An FG-GAP repeat-containing protein was identified in exoproteomic analysis of a cyanobacterial symbiont *Nostoc punctiforme* [31]. The extracellular vesicle of Fraction 1 may be a unique feature of filamentous cyanobacteria.

Acidic conditions stimulate Mg-depletion from Pchlide, resulting in protopheophorbide (Appendix A). However, it is unclear whether the accumulated protopheophorbide in the cells is selectively enriched in Fraction 1, or whether Mg-depletion occurs in part due to exposure to acidic conditions in Fraction 1. The physiological significance of the presence of two types of vesicles with such different proteins and pigment compositions is also unknown and should be investigated in future research.

The genome of *Synechocystis* sp. PCC 6803 contains six genes encoding porin isoforms, and the RNA-seq database indicated that three of these genes, which were detected in outer membrane fractions in this organism, were primarily expressed [20]. In the genome of *L. boryana*, 16 genes were discovered when BLAST search was performed using the amino acid sequence of Slr1908, the major porin isoform of *Synechocystis* sp. PCC 6803, as a query (e-value less than 4e^−45^). Only four porin isoforms were detected in Fractions 0 and 2 (Table 1), indicating that at least four of the 16 are significantly expressed and localized to EVs in dark-grown YFC2 cells of *L. boryana*. This is supported by our preliminary RNA-seq data, in which the four genes are included in the top five genes for porin isoforms in transcript abundance among the 16 genes in wild type (dg5)-grown photosynthetically.

In the photosynthetic bacterium *Rhodobacter capsulatus*, mutants deficient in bacteriochlorophyll biosynthesis, such as ZY5 (∆*bchL*), also secrete biosynthetic intermediates, including Pchlide, into the culture media [32,33]. These intermediates have been reported to be commonly bound to porin [34]. Given the lack of understanding of EVs at the time and our investigation of Pchlide secretion in cyanobacteria via EVs, it is proposed that bacteriochlorophyll biosynthetic intermediates in *R. capsulatus* are also secreted into the culture media via EVs, with porin as the key component protein.

EVs are widely distributed in most organisms and play various roles in several processes, including the recovery of nutrients (iron ions from the extracellular environments), pathogenicity through toxic compounds efflux, and gene fragments exchange [17,18]. Cyanobacterial EV production was also reported in various species, such as *Synechocystis* sp. PCC 6803 [19,20], *C. raciborskii* [28], *Prochlorococcus* sp. [35], and the cyanobacterial symbiont of *Azolla microphylla* [36]. The proposed functions of EVs in cyanobacteria include defense against cyanophages by adsorbing cyanophages, nutrient supply to heterotrophic bacteria in the food chain of marine ecosystems, horizontal gene transfer, response to stress such as UV, and intercellular communication. From an applied perspective, cyanobacterial EVs are also expected to be useful in promoting angiogenesis and wound healing [37].

In this study, we discovered that EVs contribute to the secretion of Pchlide, which accumulates abnormally in cells. This is a new role of EVs in the efflux of accumulated Chl biosynthetic intermediates, such as Pchlide as well as free Chl, which are potent photosensitizers that are extremely dangerous to cells. For this reason, Chl biosynthesis is finely regulated at multiple levels to prevent the accumulation of intermediates [3,4,5,6]. The prolamellar bodies present in etioplasts of angiosperms are thought to be one of the regulation mechanisms against Pchlide accumulation in the dark [8]. Pchlide secretion via EVs observed in YFC2 could be regarded as a defense mechanism against Chl intermediate accumulation in cyanobacteria.

The mechanism by which accumulated Pchlide in cells is loaded into EVs remains to be investigated. One of the proposed EV formation mechanisms is that EVs are formed where the interaction between the outer membrane and its underlying peptidoglycan layer is weakened. Another mechanism of EV formation is that an increase in “periplasmic pressure” due to the accumulation of misfolded proteins and peptidoglycan fragments in the periplasmic region enhances EV formation [38]. This is supported by the hypervesiculation phenotype of mutants in which genes for outer membrane-localized proteins (such as TolC) are disrupted [39]. As the biosynthetic enzymes of Chl are localized in the stroma (the cytosol in cyanobacteria) and the thylakoid membranes [4], intermediates should be accumulated in the thylakoid membrane first when a biosynthetic pathway enzyme, such as DPOR, is dysfunctional. In dark-grown YFC2 cells, Pchlide first accumulates abnormally in the thylakoid membrane. Then, Pchlide could be transported across the plasma membrane to the periplasm in some specific or non-specific transporters, thereby increasing the “periplasmic pressure” and resulting in EV formation (Figure 9). Alternatively, the accumulation of Pchlide in the periplasm may weaken the interaction between the outer membrane and peptidoglycan layer, thus stimulating EV formation.

The molecular mechanism by which the pigments accumulated in the thylakoid membrane are loaded into EVs for secretion into the culture medium is a future challenge. Furthermore, we are currently investigating whether EVs are generated in wild-type cells of *L. boryana*.

## 4. Materials and Methods

### 4.1. Cyanobacterial Strains and Growth Conditions

The filamentous and nonheterocystous cyanobacterium *Leptolyngbya boryana* strain dg5 was used as the wild type. YFC2 is a *chlL*-disrupted mutant that carries the kanamycin resistance cartridge in the coding region of *chlL*. YFC2 was cultivated in BG-11 medium [40] supplemented with 20 mM HEPES-KOH; pH7.4 [12] and 30 mM glucose (BG-11+Glc). For the dark growth, a 100-mL liquid culture in an Erlenmeyer flask was incubated in the dark with reciprocal shaking at 106 rpm (Bio-Shaker BR-3000-LF, TAITEC, Koshigaya, Japan).

### 4.2. Preparation of Pigments in the Culture Media

YFC2 was pre-cultured on a BG-11+Glc agar plate supplemented with 15 µg/mL kanamycin sulfate (Bacto Agar, Becton, Dickinson and Company, Franklin Lakes, NJ, USA) at 30 °C for seven days under continuous light (20 μmol_photon_ m^−2^ s^−1^, fluorescent lamp FL15EX-N-A, Hitachi, Tokyo, Japan), and repeated twice. For the main culture, the cells were suspended in 100 mL of BG-11+Glc liquid medium (a 500-mL Sakaguchi flask) from the agar plate of the pre-culture at an OD_730_ of 0.5 and incubated at 30 °C in the dark for seven days with reciprocal shaking (106 rpm). The supernatant was collected after centrifuging the culture (6000 rpm, 10 min, 4 °C, R10A2 rotor, Himac centrifuge CR21F, Hitachi). Pigments in the supernatant were prepared as previously reported [13]. Briefly, an equal volume of diethyl ether was added to the collected supernatant, and the upper phase was collected using a separatory funnel. The collected upper phase was allowed to stand at –20 °C before the water was extracted as ice. The recovered diethyl ether was evaporated under a stream of nitrogen gas to dry out. This was the final pigment sample in the culture medium.

### 4.3. Preparation of Pchlide Fractions from Culture Media Using Sucrose Density Gradient Centrifugation

The supernatant of the culture medium was collected by centrifugation as shown above and then centrifuged again (6000 rpm, 10 min, 4 °C, R10A2 rotor) to eliminate as many contaminated cells as possible. Then, the supernatant solution was then subjected to ultracentrifugation (45,000 rpm, 1 h, 4 °C; S50A rotor, Himac Ultracentrifuge CS100GXII, Hitachi-Koki), and the resulting precipitate was suspended in 1 mL of phosphate-buffered saline (PBS) containing 1 mM EDTA (all subsequent PBS buffers contain 1 mM EDTA). This suspension was agitated for 10 min (MicroMixer E-36, TAITEC), centrifuged at low-speed (3000 rpm, 5 min, 4 °C; ARO 15–24 rotor, centrifuge MX-301, Tommy Seiko, Tokyo, Japan), and the supernatant was collected as “Fraction 0” and stored at 4 °C in the dark.

PBS buffer containing 2.5 M, 1.6 M, and 0.6 M of sucrose was layered in 10 mL tubes (40 PA tubes, Hitachi), starting from the bottom, and finally 1 mL of the “Fraction 0” was layered, followed by ultracentrifugation (25,000 rpm, 24 h, SRP28SA rotor, Himac ultracentrifuge CP65β, Hitachi), and the yellow band of the interface between 0.6 M and 1.6 M and the green band of the interface between 1.6 M and 2.5 M were collected as “Fraction 1” and “Fraction 2”, respectively (Figure 4A, inset).

To increase the sample concentration for protein analysis, each sample was precipitated by ultracentrifugation (45,000 rpm, 1 h, 4 °C, S55A2 rotor, Himac CS100GXII) again and suspended in 100 μL of PBS buffer (10 min, MicroMixer E-36).

### 4.4. Pigment Analysis of the Culture Media

For the cell fraction (Fractions 0, 1, and 2), 900 μL of methanol was added to 100 μL of the sample, and the samples were kept on ice for 30 min. Each sample was then centrifuged (15,000 rpm, 4 °C, 10 min; ARO15-24 rotor, centrifuge MX-301), and the supernatant was analyzed using HPLC [41] (4.6 × 150 mm Symmetry C8 3.5 μm column, Waters; Shimadzu LC series, Shimadzu, Kyoto, Japan). The eluted pigments were detected by absorption at 440 nm and 690 nm (SPD-20AV, Shimadzu), and the absorption spectra of the pigments were monitored continuously by a photodiode array detector (SPD-M20A, Shimadzu). The absorption spectra of the pigment samples were measured on a spectrophotometer (model V-650, Jasco, Hachioji, Japan) before HPLC analysis. Chl(ide) and Pchlide concentrations were calculated from the absorbance at 665 nm and 640 nm using the formula in [21].

### 4.5. Preparation of Membrane Fractions from L. boryana Cells

The membrane fractions from dg5 cells were according to a previous study [11]. *L. boryana* dg5 cells were grown in the dark under the above conditions. The cells were collected using centrifugation at 6000 rpm for 10 min at 4 °C (R10A2 rotor, Himac CR21F) and were washed using PBS buffer. Then, glass beads were added to the sample tubes and the cells were crushed using a bead beater (at LA intensity, BugCrasher GM-01, TAITEC) for 1 h. After that, unbroken cells were removed using centrifugation at 2000 rpm for 3 min at 4 °C (ARO 15-24 rotor, MX-301). The supernatant was loaded onto a sucrose density gradient (2.5 M, 1.6 M, and 0.6 M sucrose in PBS buffer) and centrifuged at 25,000 rpm for 20 h at 4 °C (SRP28SA rotor, Himac CP65β). The plasma and thylakoid membranes were collected from the 0.6-M sucrose layer and the interface between the 1.6-M and 2.5-M sucrose layers, respectively. Each membrane fraction was subjected to threefold dilution using PBS buffer. Each fraction was suspended in a small volume of PBS buffer after centrifugation at 45,000 rpm for 1 h at 4 °C (S50A rotor, Himac CS100GXII).

### 4.6. SDS-PAGE and Western Blot Analysis

Furthermore, 15 μL of 2 × SDS-sample buffer containing 10% β-mercaptoethanol was added to 15 μL SDS-PAGE sample, stirred for 10 min (MicroMixer E-36), and then heat-treated (95 °C, 5 min). For mass spectrometry analysis after SDS-PAGE, acrylamide solution was added to 1% final concentration after heat treatment and stirred for 10 min. These samples were loaded onto a polyacrylamide gel (15% acrylamide) and electrophoresed at a constant voltage of 200 V for 60 min. After electrophoresis, the gels were stained with CBB (Quick CBB Plus, Wako Pure Chemicals, Osaka, Japan).

Western blotting analysis was conducted essentially as described in a study by [12]. Protein concentrations of each fraction were determined using a Protein Assay Kit (Bio-Rad, Hercules, CA, USA) with bovine serum albumin as the standard, and 0.4 µg of protein was loaded into each lane. A PVDF membrane (Immobilon P, Merck, Darmstadt, Germany) was incubated with the antisera (for anti-CmpA [11] and anti-CP47 [21] antisera), followed by the incubation with goat anti-rabbit IgG–horseradish peroxidase conjugate (Bio-Rad). The specific protein bands were visualized using an iBright 1500 (Invitrogen Thermo Fisher Scientific, Waltham, MA, USA).

### 4.7. Mass Spectroscopic Analysis

After SDS-PAGE and CBB staining, the bands were cut into strips and washed with 1 mL of 30% acetonitrile (ACN)-0.1M Trimethylammonium bicarbonate buffer (TEAB) for 15–30 min using a rotary mixer to decolorize completely. The decolorized gel pieces were dried under reduced pressure for 1 h in a VC-360 centrifugal dryer. The dried gel pieces were soaked with 30 μL of 10 ng/μL Trypsin/Lys-C Mix (Promega, Madison, WI, USA)-0.01% ProteaseMAX^TM^ surfactant (Promega)-50 mM TEAB to absorb trypsin and Lys-C into the gel. After the trypsin solution was completely absorbed into the gel, 100 μL of 0.01% proteaseMAX^TM^ surfactant-50 mM TEAB was added to the gel pieces and allowed to incubate at 50 °C for 1 h. The gel pieces were centrifuged at 12,000× *g* for 1 min and the eluate was collected. The remaining gel pieces were incubated with 500 µL of 50 mM TEAB at 50 °C for 10 min. The eluate was collected by centrifugation at 12,000× *g* for 2 min. All the eluates were mixed. The tryptic peptides were prepared using MonoSpin^®^ C18 (GL Sciences, Tokyo, Japan). Peptides were eluted with 50 μL of 70% CAN-0.1% formic acid (FA). The eluted peptide solution was dried in a centrifugal dryer and then suspended in 20 μL of 0.1% FA-2% ACN-0.001% PEG 20,000. PEG 20,000 was added to prevent the peptides from adsorbing to the tubes and vials [42].

Qualitative analysis of the peptide samples was performed by LC/MS/MS analysis using a mass spectrometer (QTRAP5500, Sciex, Tokyo, Japan) and nanoLC chromatography (Eksigent LC425, Sciex). The tryptic sample was injected into a trap column (200 μm × 0.5 mm ChromXP C18-CL) and passed through an analysis column (75 μm × 15 cm ChromXP C18-CL) and then introduced into the mass spectrometer. Furthermore, 0.1% FA and 2% ACN were used for solution A, and 0.1% FA and 100% ACN for solution B. The flow rate was 300 nL/min. The gradient was set at 0–30 min 2–35% B, 30–35 min 70% B, and 35–45 min 2% B. For qualitative analysis, information-dependent acquisition results were analyzed using ProteinPilotTM Software 5.0 (Sciex).

### 4.8. Transmission Electron Microscopy

The cells (dg5 and YFC2) used for observation were grown in the dark for 7 days. After ultracentrifugation, the pigments fraction was suspended in a small volume of PBS buffer. After mixing with 2% agarose (1:1), the samples were frozen in a high-pressure freezer (EM PACT2, Leica Microsystems, Germany) at 20–21 MPa, with flat specimen carriers (1.25 mm diameter, 200 µm depth). The samples were transferred to 2% (*w*/*v*) osmium tetroxide in acetone at −85 °C for 8 h, and kept at −80 °C for 2 days, at −30 °C for 16 h, at 4 °C for 2 h, and at room temperature for 1 h. After the subsequent freeze substitution, the samples were washed using acetone and were slowly infiltrated in a stepwise manner with increasing concentrations of resin (Agar Low Viscosity Resin, Agar Scientific, Stansted, UK) at steps 5, 10, 20, 30, 50, 75, and 100%. After degassing, the resin was cured via heating at 60 °C for 24 h. The samples were cut into ultrathin sections (90 nm thickness) using a diamond knife (Diatome) attached to an Ultra-Microtome (EM UC6, Leica), and the sections were placed on a copper grid (150 mesh, Nisshin EM). The sections were then stained using 2% (*w*/*v*) aqueous uranyl acetate for 15 min and were lead stained for electron microscopy (Sigma-Aldrich, St. Louis, MO, USA) for 4 min. The stained sections were observed using a transmission electron microscope (H-7500, Hitachi), and the images were captured using a CCD camera (Advanced Microscopy Technique, Danvers, MA, USA). The area value (nm^2^) of EV was calculated using Image J. The diameter of the circle corresponding to the area is calculated.

## Figures and Tables

**Figure 1 plants-11-00910-f001:**
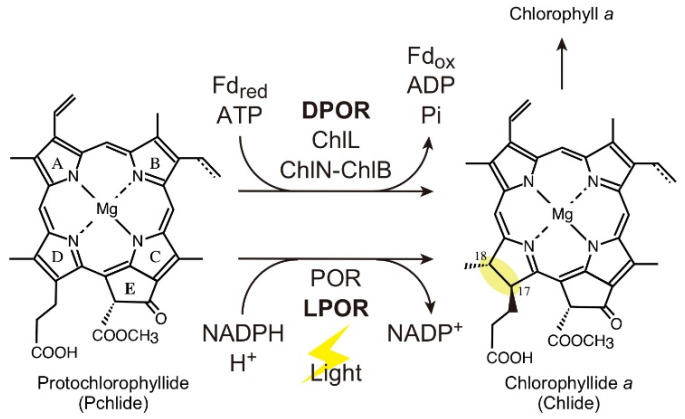
Pchlide reduction in the penultimate step of Chl *a* biosynthesis. In cyanobacteria, Pchlide is reduced by two evolutionarily unrelated enzymes, DPOR and LPOR. DPOR is a nitrogenase-like enzyme consisting of L-protein (ChlL dimer) and NB-protein (ChlN-ChlB heterotetramer). DPOR requires reduced ferredoxin (Fd) and ATP for catalysis. LPOR is a single polypeptide enzyme that uses NADPH as the reductant. Light is required for the catalysis of LPOR.

**Figure 2 plants-11-00910-f002:**
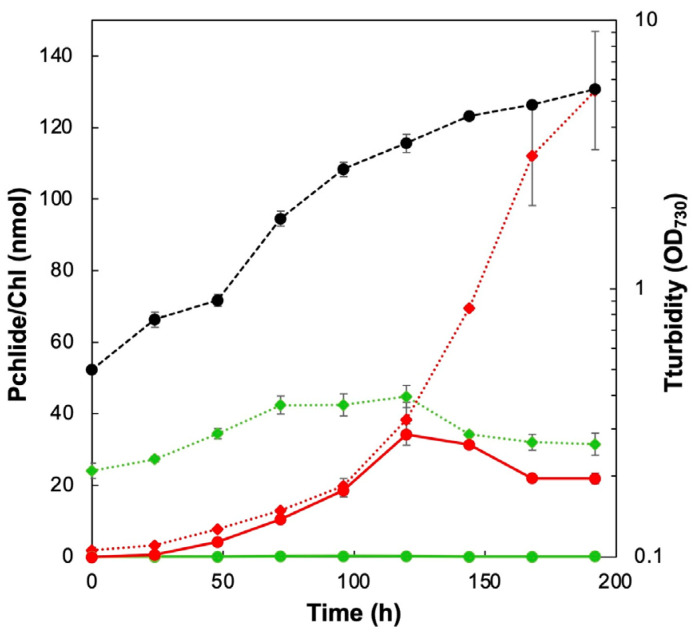
Pchlide accumulation of YFC2 during heterotrophic growth in the dark. Filled black circles indicate the turbidity (OD_730_). Chl (green) and Pchlide (red) contents in the culture media (circles, solid lines) and cells (diamonds, dotted lines) in 10 mL.

**Figure 3 plants-11-00910-f003:**
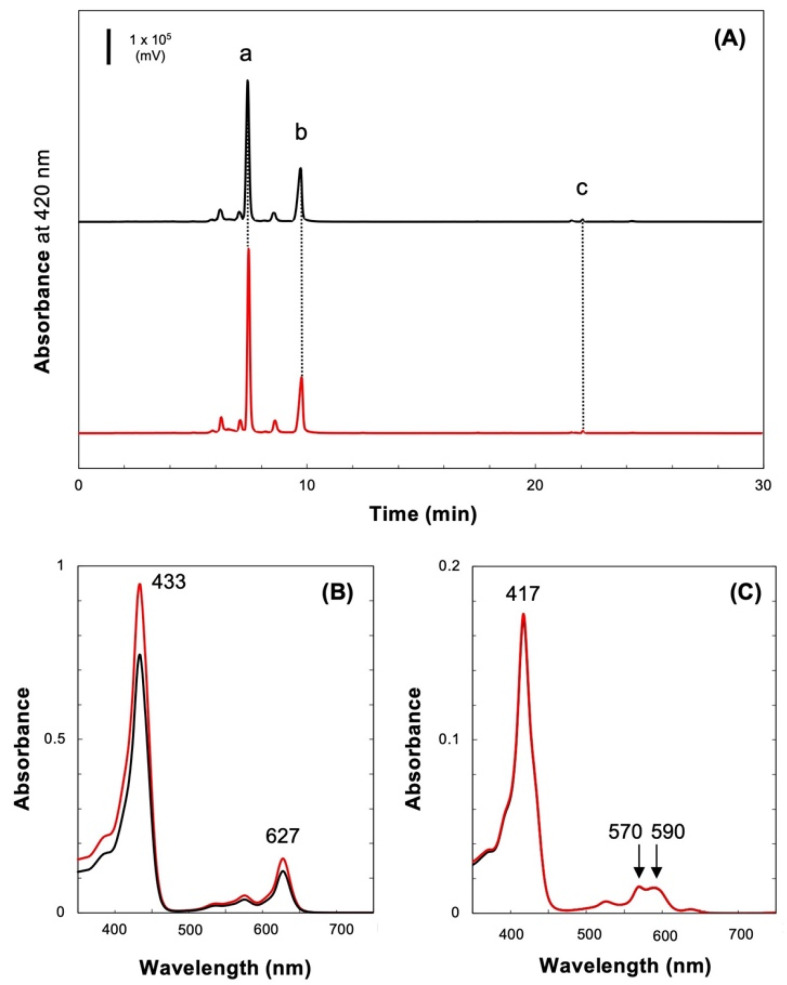
Pigment analysis of dark-grown YFC2. (**A**) HPLC profiles of pigments (90% methanol) extracted from the culture medium (black traces) and the ultracentrifugation fraction (Fraction 0, red traces). Peaks a, b, and c are monovinyl Pchlide, protopheophorbide, and Chl, respectively. (**B**) Absorption spectra of peak a of (**A**). (**C**) Absorption spectra of peak b of (**A**).

**Figure 4 plants-11-00910-f004:**
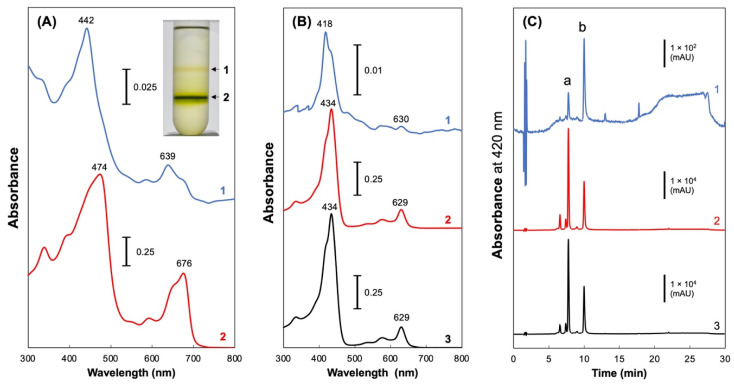
Separation of pigment fractions using sucrose density gradient ultracentrifugation. After ultracentrifugation ((**A**), inset), two bands were collected as Fractions 1 and 2, and their absorption spectra were recorded ((**A**), traces 1 and 2, respectively). Absorption spectra and HPLC profiles of the methanol extracts of the Fractions 1, 2, and 0, are shown in traces 1, 2, and 3, respectively ((**B**,**C**), respectively). Peaks a and b indicate the elution of Pchlide and protopheophorbide, respectively.

**Figure 5 plants-11-00910-f005:**
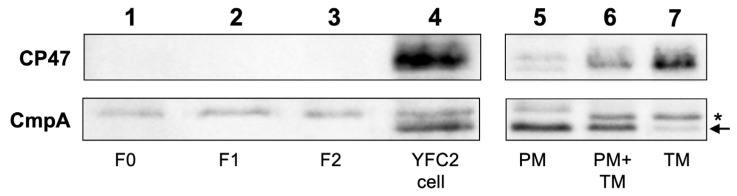
Western blotting analysis of Fractions 0, 1, and 2. Fractions 0 (lane 1), 1 (lane 2), 2 (lane 3), and a total extract of dark-grown YFC2 cells (lane 4). As controls of cross-reactivity of the antibodies, plasma (lane 5) and thylakoid (lane 7) membranes and a mixture of these membranes (lane 6) from dg5 (wild type) grown in the dark were also analyzed (0.4 µg of protein per lane). Antisera against CP47 and CmpA were used to detect thylakoid and plasma membrane-specific proteins, respectively. The CmpA signal is indicated by the arrow. * indicates a non-specific signal of the anti-CmpA antiserum.

**Figure 6 plants-11-00910-f006:**
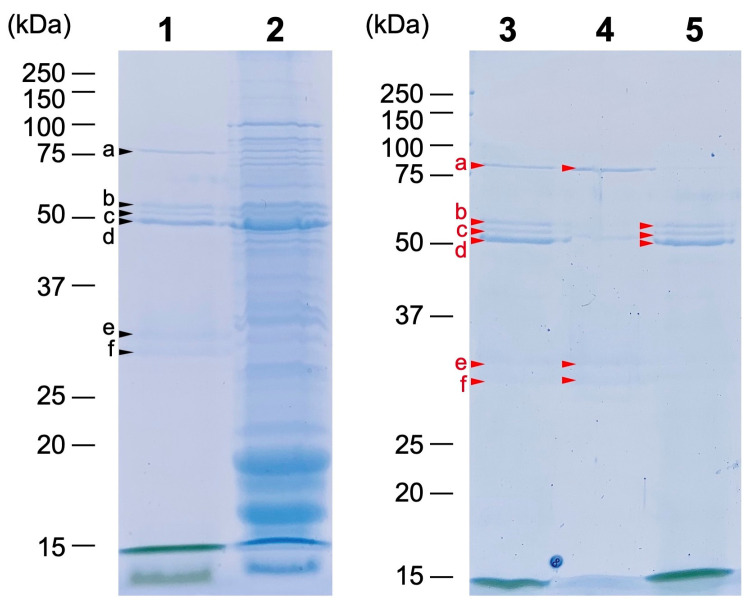
SDS-PAGE of Fractions 0, 1, and 2. Fractions 0 (lanes 1 and 3), 1 (lane 4), and 2 (lane 5) were analyzed using SDS-PAGE (CBB staining). The 6 major bands are indicated by arrowheads as a through f, from highest to lowest molecular mass. Furthermore, the total cell extract of dark-grown YFC2 was analyzed as a reference (lane 2). The bands excised for mass spectrometric analysis are indicated by small triangles.

**Figure 7 plants-11-00910-f007:**
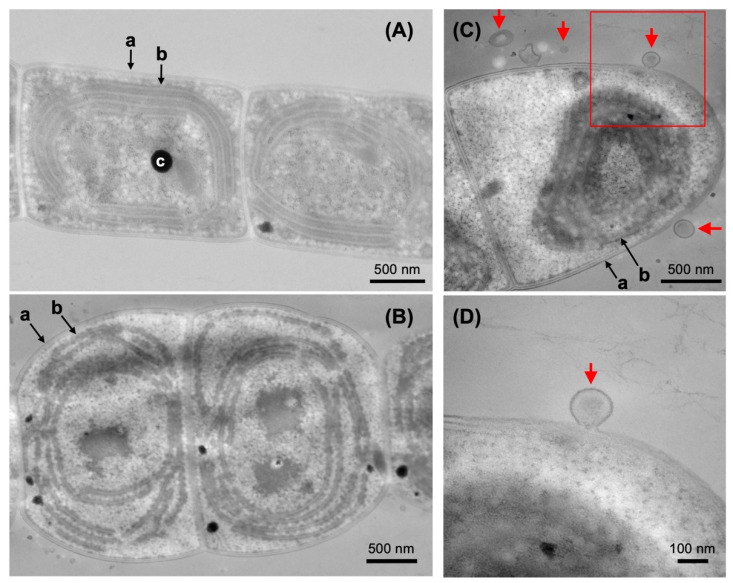
TEM observation of dg5 (wild type) (**A**) and YFC2 (**B**–**D**) cells grown in the dark. Photo (**D**) is enlarged images of photo (**C**) (red square). EV-like structures are indicated by red arrows. a, outer membrane; b, thylakoid membrane; c, lipid granule.

**Figure 8 plants-11-00910-f008:**
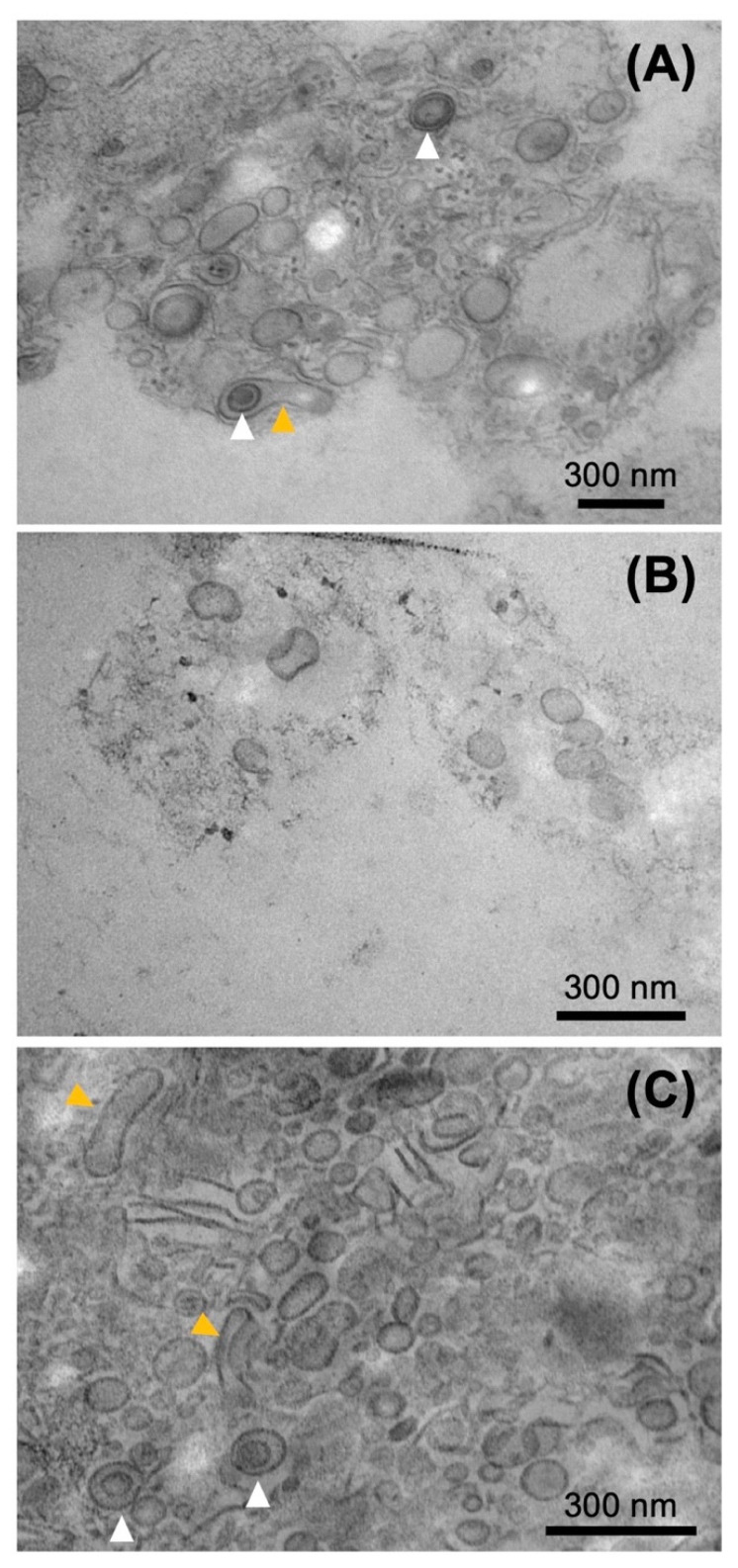
TEM observation of Fractions 0 (**A**), 1 (**B**), and 2 (**C**). Multi-lamellar and pear-shaped EVs are shown by white and orange arrowheads, respectively.

**Figure 9 plants-11-00910-f009:**
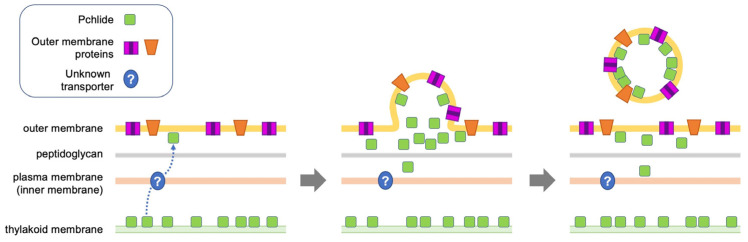
Model of EV formation in YFC2 cells. Pchlide molecules accumulated in the thylakoid membrane are transported to the periplasm by the action of an unknown transporter located in the plasma membrane. The abnormal accumulation of Pchlide in the periplasm may cause an increase in the periplasm pressure or weaken the interaction between the outer membrane and peptidoglycan layer, thus stimulating EV formation.

**Table 1 plants-11-00910-t001:** Protein identification of the fractions from sucrose density gradient centrifugation.

Fraction	Band ^1^	Peptides ^2^	Accession Number	Annotation	Locus Tag ^3^	Predicted Mol. Mass (kDa)
0	a	26	WP_017290631.1	FG-GAP repeat-containing protein	07530	80.4
	b	19	WP_017287236.1	Carbohydrate-selective porin, OprB family	40860	58.4
		11	WP_017287165.1	Glutamine synthetase (GlnA)	41490	65.3
		11	WP_144056229.1	Outer membrane efflux protein (TolC)	23750	52.8
	c	41	WP_017288757.1	Cyanobacterial porin	25860	61.4
		10	WP_017287236.1	Carbohydrate-selective porin	40860	58.4
		9	WP_017287123.1	Carbohydrate-selective porin, OprB family	41910	58.4
	d	21	WP_017287123.1	Carbohydrate-selective porin, OprB family	41910	58.4
		10	WP_017288757.1	Cyanobacterial porin	25860	61.4
		8	WP_026148594.1	Carbohydrate-selective porin, OprB family	34630	60.8
1	a	17	WP_017290631.1	FG-GAP repeat-containing protein	07530	80.4
2	b	10	WP_017287236.1	Carbohydrate-selective porin, OprB family	40860	58.4
	c	15	WP_017288757.1	Cyanobacterial porin	25860	61.4
	d	10	WP_017287123.1	Carbohydrate-selective porin, OprB family	41910	58.4

^1^ Letters a to f are corresponding to the band letters in Figure 6; ^2^ Number of peptide fragments detected with >95% accuracy (Appendix A); ^3^ LBDG_xxxxx.

## Data Availability

Data are contained within the article or Appendix A.

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
