# Peer review of "Extracellular Vesicle-Mediated Secretion of Protochlorophyllide in the Cyanobacterium Leptolyngbya boryana"

_plants, 2022, doi:10.3390/plants11070910_

Round 1

Reviewer 1 Report

The study of extracellular vesicles (EV) as vehicle proteins and molecules in the plant cell communication is a novel topic with great interest. In this work, the characterization of EV containing Pchlide resulted of interest. A new role of EVs in the efflux of accumulated Chl biosynthetic intermediates, such as Pchlide, as a mechanisms to protect cells is proposed.

However, some metodological questions must be solved;

The percentage of other membranes contamination must be provided in the separated fractions. For that, enzymatic activities assays for each type of membrane can be carried out.

The size of EV can be provided from electronic microscope images in order to characterize the EV vesicles.

Finally, in the discussion more details about the role or the type of proetins found in EV in photobacterial or other photosynthetic organism can be provided

Author Response

Dear Reviewer 1,

We appreciate the helpful suggestions offered by Reviewer 1, as the comments are very useful for improving this manuscript.

Following the reviewer 1's comments, we write our responses to the comments after >.

Reviewer 1: The study of extracellular vesicles (EV) as vehicle proteins and molecules in the plant cell communication is a novel topic with great interest. In this work, the characterization of EV containing Pchlide resulted of interest. A new role of EVs in the efflux of accumulated Chl biosynthetic intermediates, such as Pchlide, as a mechanisms to protect cells is proposed.

However, some metodological questions must be solved;

The percentage of other membranes contamination must be provided in the separated fractions. For that, enzymatic activities assays for each type of membrane can be carried out.

The size of EV can be provided from electronic microscope images in order to characterize the EV vesicles.

Finally, in the discussion more details about the role or the type of proetins found in EV in photobacterial or other photosynthetic organism can be provided

>Thank you for your positive response to this manuscript.

1) The percentage of other membranes contamination must be provided in the separated fractions. For that, enzymatic activities assays for each type of membrane can be carried out.

> We carried out Western blot analysis using antisera against CP47 and CmpA that are localized in thylakoid and cytoplasmic membranes, respectively (Figure 5). No signals reacting to either antiserum were detected in the protochlorophyllide (Pchlide) fractions. This is consistent with almost no Chl peaks in absorption spectra of Methanol extact and HPLC profiles (Figure 4B, C).

2) The size of EV can be provided from electronic microscope images in order to characterize the EV vesicles.

> We measured size of EVs observed in the vicinity of YFC2 cells and Fractions 0, 1, and 2, and showed the result as histograms (Figure S3).

3) Finally, in the discussion more details about the role or the type of proetins found in EV in photobacterial or other photosynthetic organism can be provided

We added sentences describing specific functions of cyanobacterial EVs just after cyanobacterial species that were reported producing EVs (Discussion). In addition, to introduce the possibility that outer membrane proteins could be involved in EV formation, we additionally describe a paper reporting that mutants deficient in the gene encoding the TolC protein, which localizes to the outer membrane, exhibit hypervesiculation phenotype.

Sincerely yours,

Dr. Yuichi Fujita

Professor

Graduate School of Bioagricultural Sciences

Nagoya University

Furo-cho, Chikusa-ku, Nagoya 464-8601

Tel. +81-(0)52-789-4089; Fax. +81-(0)52-789-4091

E-mail: fujita@agr.nagoya-u.ac.jp

Reviewer 2 Report

The authors describe an observation of extracellular vesicles of a DPOR-deficient mutant (YFC2) of the cyanobacterium Leptolyngbya boryana, which are proposed to contain the substantial amounts of protochlorophyllide. This observation was not performed in the wild-type strain, as suggested and indicated in the title. The report verifies accumulation of protochlorophyllide (Pchlide) and protopheophorbide. However, it was not clarified whether latter tetrapyrrole intermediate was found due to the extraction procedure or as result of deregulation in tetrapyrrole metabolism of the yfc2 mutant or the segregation in the outer membrane. In addition, certain proteins were detected in the purified fractions form the media, which contain also both tetrapyrrole, but mainly protochlorophyllide.

This manuscript deals mainly with these observations and descriptions without any detailed and confirmative explorations. A physiological relevance of PChlide secretion by extra cellular vesicles was not investigated and only discussed at the end of the manuscript. The suggestions and indications of these preliminary findings could certainly be substantiated.

An explanation should be given that inside the cells PChlide content can increase, while the PChlide content in the culture media increases only slightly and remains – at its best – stable over the entire time frame of the experiment (Figure 2). Increasing amounts of secreted PChlide are expected.

In addition, it is not obvious that protochlorophyllide isolated from media is derived from the extracellular vesicles mentioned in this manuscript. It is not obvious that fraction 0 consists of the described vesicles (EV). The same holds true for the fraction 1 and 2 which are derived from fraction 0 after sucrose density gradient ultracentrifugation. Moreover, no obvious experimental indication was provided if the proteins summarized in Table 1 have any association to Pchlide. It is not obvious whether these proteins are the dominant protein candidates which were obtained by mass spectrometry.

Regrettably, the entire manuscript prefers more suggestions and subtle indications rather than experimental proofs of the PChlide–containing vesicles and the Pchlide-binding proteins in the isolated fraction (see line 141”… proteins of the fraction that interacted with PChlide…) or (line 257… free Chl…)

Author Response

Dear Reviewer 2,

We appreciate the helpful suggestions offered by reviewer 2, as the comments are very useful for improving this manuscript.

Following the reviewer 2's comments, we write our responses to the comments after >.

Reviewer 2: The authors describe an observation of extracellular vesicles of a DPOR-deficient mutant (YFC2) of the cyanobacterium Leptolyngbya boryana, which are proposed to contain the substantial amounts of protochlorophyllide. This observation was not performed in the wild-type strain, as suggested and indicated in the title. The report verifies accumulation of protochlorophyllide (Pchlide) and protopheophorbide. However, it was not clarified whether latter tetrapyrrole intermediate was found due to the extraction procedure or as result of deregulation in tetrapyrrole metabolism of the yfc2 mutant or the segregation in the outer membrane. In addition, certain proteins were detected in the purified fractions form the media, which contain also both tetrapyrrole, but mainly protochlorophyllide.

This manuscript deals mainly with these observations and descriptions without any detailed and confirmative explorations. A physiological relevance of PChlide secretion by extra cellular vesicles was not investigated and only discussed at the end of the manuscript. The suggestions and indications of these preliminary findings could certainly be substantiated.

An explanation should be given that inside the cells PChlide content can increase, while the PChlide content in the culture media increases only slightly and remains – at its best – stable over the entire time frame of the experiment (Figure 2). Increasing amounts of secreted PChlide are expected.

In addition, it is not obvious that protochlorophyllide isolated from media is derived from the extracellular vesicles mentioned in this manuscript. It is not obvious that fraction 0 consists of the described vesicles (EV). The same holds true for the fraction 1 and 2 which are derived from fraction 0 after sucrose density gradient ultracentrifugation. Moreover, no obvious experimental indication was provided if the proteins summarized in Table 1 have any association to Pchlide. It is not obvious whether these proteins are the dominant protein candidates which were obtained by mass spectrometry.

>Thank you for your critical responses to this manuscript.

We focus on the secretion of Pchlide in a chlL-deficient mutant YFC2 growing heterotrophically in the dark. For this reason, this manuscript consists mainly of observations. Experimental verification and research on the physiological significance of this phenomenon will be the subject of future work. Reviewer 2's criticism is a point against the style of this manuscript itself, which requires many more months of experimentation to fully meet the reviewer 2’s requirements. Furthermore, incorporating those results would equate to writing a new paper, not a revision. Due to time constraints, this revision was limited to some additional experiments and revisions of descriptions that could be addressed within one month.

The most important point is to confirm whether Fractions 0, 1 and 2 are really composed of EVs, as pointed out by reviewer 2. We succeeded in observing EV-like structures in all fractions by using a new high-pressure freezing method to prepare samples for electron microscopy. The obtained photographs are shown in Figure 8. Since this sample preparation method allowed us to observe cyanobacterial cells in more detail, we replaced the original electron micrographs with the new ones as shown in Figure 7.

In the time course of intracellular and extracellular Pchlide contents during heterotrophic growth of YFC2 (Fig. 2), the culture medium and cells were separated by low-speed centrifugation, and the respective pigment contents were quantified. We recognized that after 120 h the precipitates from low-speed centrifugation contained pigment aggregates that were clearly different from the cells, and we briefly mentioned it in Discussion. However. Reviewer 2 pointed out that the results in Figure 2 are contrary to the expectation that the Pchlide content of the culture medium should also be increased. In response to this point, we performed absorption spectra, HPLC analysis, and SDS-PAGE analysis on the Pchlide aggregates. We found that the intracellular Pchlide contents were overestimated because the pellets of low-speed centrifugation contained not only cells but also Pchlide aggregates. This point was added to Discussion and the results are shown in Figure S5.

In addition, we added a new model (Fig. 9) in Discussion regarding the relationship between EV formation and Pchlide accumulation to propose the possibility that Pchlide accumulation promotes EV formation. The hypothesis is that EV formation is stimulated by Pchlide accumulation in the periplasm. This idea provides a new perspective on the EV formation mechanism proposed so far: A decrease in the interaction between the outer membrane and the peptidoglycan layer or an increase in periplasmic pressure due to accumulation of unfolded proteins in the periplasm stimulates EV formation. We hypothesize that Pchlide accumulation in the periplasm promotes EV formation, either by decreasing the interaction between the outer membrane and the peptidoglycan layer or by increasing periplasmic pressure. We believe that the proposal provides the basis for future research.

In detection by mass spectrometry, proteins with higher content tend to be detected mainly. In this mass spectrometry analysis, the list of proteins in Table 1 can be regarded as the major proteins in each Fraction (i.e., EVs) because the values of molecular mass of the detected proteins is in good agreement with those of the major bands in CBB staining.

The ratios of Pchlide to protopheophorbide in Fraction 0 and 2 were almost identical to that in methanol extract from cells, suggesting that protopheophorbide exists in the Pchlide accumulating YFC2 cells. Methanol extraction is usually used to determine the pigment composition of cells. This suggests that protopheophorbide is generated in Pchlide accumulating YFC2 cells. The reason why protopheophorbide accumulates more in Fraction 1 than other Fractions is still unknown at this time.

Sincerely yours,

Dr. Yuichi Fujita

Professor

Graduate School of Bioagricultural Sciences

Nagoya University

Furo-cho, Chikusa-ku, Nagoya 464-8601

Tel. +81-(0)52-789-4089; Fax. +81-(0)52-789-4091

E-mail: fujita@agr.nagoya-u.ac.jp